# Epidemiology of *Aeromonas* Species Bloodstream Infection in Queensland, Australia: Association with Regional and Climate Zones

**DOI:** 10.3390/microorganisms11010036

**Published:** 2022-12-22

**Authors:** Holly A. Sinclair, Felicity Edwards, Patrick N. A. Harris, Claire Heney, Kevin B. Laupland

**Affiliations:** 1Herston Infectious Diseases Institute, Brisbane, QLD 4029, Australia; 2Faculty of Health, Queensland University of Technology (QUT), Brisbane, QLD 4029, Australia; 3University of Queensland Centre for Clinical Research, Faculty of Medicine, University of Queensland, Brisbane, QLD 4029, Australia; 4Department of Microbiology, Pathology Queensland, Brisbane, QLD 4029, Australia; 5Department of Intensive Care Services, Royal Brisbane and Women’s Hospital, Brisbane, QLD 4029, Australia

**Keywords:** *Aeromonas*, bacteremia, incidence

## Abstract

*Aeromonas* species can cause severe bloodstream infection (BSI) however, few studies have examined their epidemiology in non-selected populations. The objective of this study was to describe the incidence and determinants of *Aeromonas* species BSI in Queensland, Australia. A retrospective population-based cohort study was conducted during 2000–2019. *Aeromonas* species BSI were identified by laboratory surveillance and clinical and outcome information through data linkages to statewide databases. A total of 407 incident *Aeromonas* species BSI were identified with an age- and sex-standardized incidence of 5.2 per million residents annually. No trend in annual incidence rate during two decades of surveillance was demonstrated. Significant variable monthly occurrences were observed with highest rates during warmer, wetter months, and lowest rates during winter and dry periods. There was significant variability in incidence accordingly to region and climate zones, with higher rates observed in tropical north regions and lowest in southeastern corner. The highest incidence was observed in very remote and hot areas in Queensland. Cases were infrequent in children and risk was highest in elderly and males. Seventy-eight patients died within 30 days with a case-fatality rate of 19%. Older age, non-focal infection, higher Charlson score, and monomicrobial bacteremia were independent risk factors for death. Demographic and climatic changes may increase the burden of these infections in future years.

## 1. Introduction

*Aeromonas* species are Gram-negative, facultatively anaerobic, motile, oxidase positive bacilli demonstrating typical beta-hemolysis on blood agar [1]. *Aeromonas* species are ubiquitous within aquatic environments causing severe human infection through direct water-exposure or consumption of contaminated water or food. The two major clinical presentations that have been recognized are skin and soft tissue infection that can progress to necrotizing fasciitis from water-related wounds, and biliary tract sepsis from a gastrointestinal source. Malignancy, immunosuppression, and chronic liver disease are considered risk factors for severe infection [1,2]. Although representing an infrequent cause of bacteremia burden overall, *Aeromonas* species can lead to significant case-mortality rates of one in three [2,3,4,5,6]. This is accompanied by a myriad of intrinsic antimicrobial resistance mechanisms including to carbapenems [7]. *Aeromonas* species bloodstream infection (BSI) is more commonly community-acquired, however healthcare associated infection can occur in context of trauma, burns or medicinal leech therapy [1,8].

*Aeromonas* species most frequently implicated in human infection include *A. hydrophila*, *A. veronii* bv. *sobria*, *A. caviae* and the emerging pathogen, *A. dhakensis* [9,10]. *Aeromonas* species are unable to be identified accurately to species level based on standard microbiology laboratory phenotypic tests and require more than one house-keeping gene or whole genome sequencing for definitive species identification [9,11,12,13,14,15].

*Aeromonas* species infection occurs more frequently in warmer regions and after natural disasters [1,12]. *Aeromonas* species were found in high numbers in floodwaters during Hurricane Katrina and represented the most common organism isolated in skin and soft tissue infection (22.6%) in Thailand during the tsunami in 2004 [16,17]. Locally, we see spikes of *Aeromonas* species infection during flooding in Queensland and we need to better understand the epidemiology in this region and the relationship with rainfall. The worldwide incidence of *Aeromonas* species infection is largely unknown with limited population-based studies and many case series and small single centre cohorts [4,6,18,19,20]. Incidences of *Aeromonas* species bacteremia have been reported as 0.66-, 1.5-, and 76-per million population in France, England and Wales, and Southern Taiwan, respectively [21]. However, population-based studies are lacking from other regions and studies to date have not been adequately designed to examine changes over time and within regions. The objective of this study was to describe the epidemiology of *Aeromonas* species BSI at the population level in Queensland, Australia over a twenty-year period to assess incidence, secular trends and seasonal determinants, and outcomes including mortality and risk factors associated with death.

## 2. Materials and Methods

We conducted a retrospective, population-based, laboratory surveillance cohort study of *Aeromonas* species BSI in Queensland, Australia. All incident cases of *Aeromonas* BSI were identified in the publicly funded system between 1 January 2000, and 31 December 2019. The population of Queensland, Australia in 2019 was 5.1 million persons. The publicly funded system in Queensland includes sixteen hospitals and healthcare service (HHS) regions. This study was approved by the human research ethics committee at the Royal Brisbane and Women’s Hospital (LNR/2020/QRBW/62494).

Blood cultures during the surveillance period were collected in BacT/Alert FA Plus (aerobic), FN Plus (anaerobic), and PF Plus (pediatric) medium bottles from community and institutional collection sites for Pathology Queensland. Blood culture bottles were incubated within the BacT/Alert 3D system (bioMérieux, Durham, NC, USA) until 2018 and then BacT/Alert VIRTUO system (bioMérieux) in the Pathology Queensland central referral laboratory after 2018. The Central laboratory is a referral laboratory for cultures from Greater Brisbane area and several rural Queensland sites. Blood culture bottles were incubated for five days and discarded if no growth. Bacterial identification methods included Vitek^®^2 GN ID card, API commercial tests, biochemical tests including Voges-Proskauer test, and matrix-assisted laser desorption ionization-time of flight mass spectrometry (Vitek^®^ MS) depending on the laboratory and the processes at the time. It is well described that phenotypic identification and commercial systems are inaccurate at identifying *Aeromonas* to species level [9]. Given the lack of comprehensive species allocation during the surveillance period, we did not report individual *Aeromonas* species. Antimicrobial susceptibility testing and interpretation with Clinical and Laboratory Standards Institute (CLSI) and the European Committee on Antimicrobial Susceptibility Testing (EUCAST) breakpoint criteria at the time were determined using Vitek^®^2 AST card, disc diffusion, or E-test^®^ minimum inhibitory concentration (MIC). From 2014 there was a change in reporting meropenem susceptibilities and recorded meropenem susceptibilities may not reflect the presence of carbapenemase-producing *Aeromonas* species as previously described [9].

Positive blood cultures with *Aeromonas* species were identified and then linked to statewide hospital admissions and death registries to obtain clinical and outcome data. Admissions to private or public institutions within Queensland were identified and discharge diagnostic codes (ICD-10AM) recorded. All encounters, including interhospital transfers within the state that were associated with the management of the BSI were included as the index hospitalization. The Registry of General Deaths as of 31 December 2020 was queried to confirm deaths within Queensland.

Previous validated definitions and algorithms were used to classify BSI episodes [22,23]. Definition of an incident BSI episode was the first isolation of *Aeromonas* species in a blood culture per patient. Repeat *Aeromonas* species blood cultures within 30 days were included as the same incident episode. Polymicrobial BSIs were defined as one or more organisms isolated with the *Aeromonas* species within a 48 h period [24]. Index positive blood cultures that were over two calendar days after hospital admission or within two days after discharge were classified as hospital-onset BSIs [25]. Community-onset BSIs were classified as those within two days of hospital admission or within the community. Healthcare-associated BSI were defined as those from nursing home residents, those within a healthcare institution within thirty days and/or hospital admission for more than two days within ninety days prior to the index blood culture [26]. Community-associated BSIs were those community-onset BSIs that did not fit the criteria for healthcare-associated. The Charlson Comorbidity Index was used to define the comorbidities [27,28]. The focus of infection was assigned based on the diagnosis-related group and primary diagnosis from hospital discharge codes.

Incident BSI episodes were the primary unit of analysis and were reported as age- and sex-standardized annual rates per million population (to the 2019 Queensland population). Stata 17 (StataCorp, College Station, TX, USA) was used for the data analysis. Data from the Australian Bureau of Statistics and Queensland Health was used to determine denominator data for age, sex and hospital and health service area [29]. Climate zones according to Queensland regions were obtained from the Australian Bureau of Meteorology including temperature and humidity zones, Köppen classification, and seasonal rainfall (BOM.gov.au (accessed on 19 November 2021)). Climate zones in Queensland include 1, high humidity, warm winter; 2, warm humid summer, mild winter; 3, hot dry summer, warm winter; and 5, warm temperate. Definition of the major seasonal rainfall zones included summer dominant (marked wet summer and dry winter); Summer (wet summer and low winter rainfall); and arid (low rainfall). Average annual rainfall for regions was obtained from the Australian Bureau of Meteorology. Non-Queensland residents were excluded. The annual number of total blood cultures performed at Pathology Queensland were recorded. Incidence rate ratios (IRRs) were calculated with 95% confidence intervals (CI) for group comparison and a *p*-value < 0.05 were considered statistically significant.

## 3. Results

### 3.1. Incidence and Acquisition

A total of 407 incident *Aeromonas* species BSI were identified among 400 Queensland residents (seven individuals had second incident episodes) during the two decades of surveillance. Eighty-four (20.7%) episodes were of hospital-onset, 149 (36.6%) were healthcare-associated, and 174 (42.8%) were community-associated. The overall age- and sex-standardized incidence in Queensland was 5.2 per million annually. Moderate year-to year variability was observed during the surveillance without an overall trend in the annual incidence rate over the study duration as shown in Figure 1.

### 3.2. Incidence Rates and Associations with Region, Remoteness, Climate, and Rainfall

Significant monthly variation in incidence of *Aeromonas* species BSI occurred with highest rates observed during warmer, wetter months, and lowest rates during the winter (June–August) as shown in Figure 2. Significant variability in incidence occurred according to region within Queensland, with highest rates observed in the tropical north regions and lowest in the southeastern corner as shown in Figure 3. Less month to month variation occurred in the tropical regions; regions with high humidity summer and warm winter; and summer dominant seasonal rainfall (Appendix A). More variation in monthly *Aeromonas* species BSI were seen in subtropical regions with warm humid summers and mild winters; and wet summers with low winter rainfall. Low numbers were seen in temperate regions with low average rainfall and cooler temperatures. The highest incidence was seen in very remote regions (13.0 per million population) and in the regions with the highest annual temperature regardless of the annual rainfall, including Central West Queensland (16.7 per million population) shown in Figure 4 and Appendix A. The lowest incidence occurred in major cities (3.0 per million population) compared to very remote regions (Figure 5).

### 3.3. Patient Characteristics, Source of Bloodstream Infection, Polymicrobial Infection, and Antimicrobial Susceptibilities

The median age for *Aeromonas* species BSI was 68.6 (IQR, 55.3–79.3) years and 142 (34.9%) were female. *Aeromonas* species BSI were rare in children and young adults, and the risk progressively increased with age as shown in Figure 5. Overall males were at higher risk (5.4 vs. 3.3; IRR, 1.7; 95% CI, 1.33–2.0; *p* < 0.0001), with excess risk among males related to advancing age (Figure 5). Comorbidities were common with a median Charlson score of 3 (IQR, 1–5). Only 101 (24.8%) of cases had a comorbidity score of zero. Cases with community-associated infection were more likely (*p* < 0.001) to have no comorbidities (61/174; 35.1%) as compared to healthcare-associated (26/149; 17.4%) or hospital-onset (14/84; 16.7%). An abdominal/gastrointestinal focus of infection was the most frequently identified (159; 39.1%) followed by soft tissue (29; 7.1%), lower respiratory (15; 3.7%), bone/joint (5; 1.2%), genitourinary (5; 1.2%), endovascular (3; 0.7%), and head and neck (1; 0.2%). No focus of infection was reported in 190 (46.7%) of cases.

More than one-third of incident infections (154; 37.8%) were of polymicrobial etiology with two or more isolates, and these included three species in 50 cases (8%), four in 14 (2%), five in seven (1%), six in two (< 1%), and 7 isolates in one case (<1%). Among the total 228 co-isolated organisms, the most common were *Escherichia coli* (84; 36.8%), followed by *Klebsiella pneumoniae* (32; 14.0%), *Klebsiella oxytoca* (20; 8.8%), *Enterobacter cloacae* (15; 6.6%), *Stenotrophomonas maltophilia* (5; 2.2%), and *Pseudomonas aeruginosa* (4; 1.8%). Among *Aeromonas* species isolates tested, resistance to ciprofloxacin (2/400; 0.5%), co-trimoxazole (12/398; 3.0%), gentamicin (3/398; 0.8%), and cefepime (0%) were infrequent, whereas resistance to meropenem was reported in 70/356 (19.7%) of cases.

### 3.4. Mortality and Risk Factors Associated with Death

Seventy-eight patients died within 30 days of incident *Aeromonas* species BSI with a case-fatality rate of 19.2%. The highest case-fatality was observed among episodes with no focus (53/190; 27.9%), followed by soft tissue (6/29; 20.7%), lower respiratory (3/15; 20.0%), and gastrointestinal/abdominal (16/159; 10.1%) foci. No deaths were observed among those with bone/joint, head and neck, endovascular, or genitourinary foci of infection. A logistic regression model (*n* = 400, area under receiver operator characteristic 0.7767; goodness of fit *p* = 0.93) including only first episodes of *Aeromonas* species BSI found older age, non-focal infection, higher Charlson score, and monomicrobial bacteremia independent risk factors associated with death as shown in Table 1.

## 4. Discussion

We conducted a population-based, laboratory surveillance, retrospective study on *Aeromonas* species BSI in Queensland, Australia, over a twenty-year period. This is the first study to calculate the incidence of *Aeromonas* BSI in Australia and show variations of occurrence according to region, season, climate, and rainfall.

The overall incidence of *Aeromonas* BSI we observed was 5.2 per million population annually. The first population-based study on *Aeromonas* infections was from California in 1988, and reported annual incidence of 10.6 per million population from gastrointestinal tract (81%), wound (9%) and blood cultures (5%) [30]. This equates to an incidence of 0.53 per million population of *Aeromonas* species BSI during their surveillance period. In France, a 6-month prospective nationwide study in 2006 reported ninety-nine *Aeromonas* clinical isolates with an estimated incidence of 1.62 per million population [1]. When including only cases of bacteremia, there was an estimated annual incidence of 0.66 episodes per million population [21]. In 2004, the incidence of *Aeromonas* bacteremia in England and Wales was estimated to be 1.5 per million population [29]. Taiwan has a high incidence of *Aeromonas* species BSI in comparison with 76 cases per million population [21]. This corresponded to a 143-, 50-, and 115-fold higher incidence compared to California, England/Wales, and France, respectively, and a 14.6-fold higher incidence than Queensland, Australia. Our annualized incidence of *Aeromonas* species BSI is 9.8-, 3.5- and 7.9-fold higher than California, England/Wales, and France, respectively. Suggested reasons for increased incidence in Taiwan was a population susceptible to invasive *Aeromonas* species infection including chronic liver disease and chronic hepatitis B and C; seafood markets and higher numbers of *Aeromonas* species within warm aquatic environments [21]. Queensland similarly has warmer year-round temperatures and aquatic environments favorable for *Aeromonas* species growth. Surprisingly, the incidence was high in the arid areas in central Queensland. This may be related to an unknown reservoir of *Aeromonas* species and the warm year-round temperatures.

We observed highest rates of *Aeromonas* species BSI during the warmer and wetter months of the year in regions with seasonal variation in temperature and rainfall, and geographically in the tropical north regions as compared to temperate and dry regions. The incidences of *Aeromonas* species BSI in the tropical Torres and Cape and North West regions were 12.5 and 10.1 per million population (Figure 3), respectively, which is over 3-fold higher than the incidence in the Southern more temperate climate regions including Darling Downs, South West, Metro South, and Gold Coast regions. Interestingly, no seasonal variation in occurrence of *Aeromonas* species BSI was demonstrated in a population-based study in Taiwan [21]. This is similar to our findings of less monthly variation in the number of isolates in tropical regions with higher temperatures all year round, compared to subtropical regions. A 14-year retrospective study on *Aeromonas* species BSI in Korea found similar correlation to warmer months of the year with 69% of cases occurring in warmer seasons with majority being community-acquired during these times [31]. This is important information and provides potential impact on human health with infections from *Aeromonas* species in regions with higher temperatures and the impact of climate change in the future, when combined with increasing frequency and intensity of natural disasters and flooding.

Unexpectedly and unique to this population-based study is the high incidence of *Aeromonas* species BSI (16.8 per million population) in the Central West HHS which largely comprises a low rainfall and more arid environment with very high mean temperatures year-round. Although there were only four cases observed here, the incidence was high due to its small population. The occurrence of *Aeromonas* species BSI was highest in very remote areas in Queensland (4.3-fold increase compared to major cities), a significant finding with important public health implications. Despite only 1% of the total Queensland population, very remote regions accounted for 3.5% of the *Aeromonas* species BSI. Possible reasons for this could be environmental or population based. Most importantly, we need further investigation for a source of *Aeromonas* species infection in these regions. One possibility is that rural properties in Queensland are too remote to access town water supplies and water supplies can come from rainwater, groundwater and surface water which may be contaminated with *Aeromonas* species that proliferate during very warm temperatures. We do see spikes of infection related to periods of high rainfall and flooding likely associated with increased exposure and water-related injuries, and there seemed to be a stronger association with rainfall in the subtropical regions with more seasonal variations. Climate change with warmer global temperatures could increase the incidence of *Aeromonas* species BSI over time.

Here, we found a higher rate of *Aeromonas* species BSIs to be healthcare-associated (37%) and hospital-onset (21%) compared to other studies that have documented 71 to 79% community-acquired BSI [1,2,31]. In Taiwan, monomicrobial *Aeromonas* species BSI were mostly community-acquired (74%) and those that were community-acquired, were more likely to have cirrhosis and a high severity score at onset and had poorer prognosis compared to nosocomial-acquired BSI [2]. Nosocomial *Aeromonas* species BSI occurred more frequently in those with malignancy, with lower severity score and had better prognosis but were more frequently resistant to cefotaxime [2]. It was suggested that nosocomial *Aeromonas* BSIs may be related to colonization within the gastrointestinal tract resulting in bacteremia under antibiotic selective pressure during hospitalization rather than acquired from the hospital environment [2].

The focus of infection was most commonly from a gastrointestinal source (39%) and no focus was documented in 47% of cases. Only 7% were attributed to a soft tissue source which was lower than expected. The two major sources of invasive *Aeromonas* infection are thought to be from biliary tract (gastrointestinal source) and skin and soft tissue infection [9]. In this study, the skin and soft tissue group may be under-represented and may be part of the group with no focus reported. However, similarly, in Korea the major cause of bacteremia over a fourteen-year period was hepatobiliary tract infection (50.6%), peritonitis (18.5%), and primary bacteremia (17.9%), with only 4.2% attributed to SSTI [31]. Wu et al. [6] demonstrated similar rates of undefined source of bacteremia (58.2%) in a cohort of monomicrobial *Aeromonas* species BSI.

In this study, the overall 30 days case-fatality rate in patients with *Aeromonas* species BSI was 19%. This compares to 30 days case-fatality for Enterobacterales bacteremia in Australia in 2019 of 11.8% [32]. We found the risk of death significantly higher in cases where no focus was documented, in older individuals, higher Charlson score and monomicrobial BSI. It remains unexplained as to why polymicrobial infections were at lower case-fatality risk. The highest case-fatality rate was among those with no focus (28%), followed by soft tissue (21%) and gastrointestinal (10%). This signifies the importance of rapid identification of the source of infection and prompt source control to improve the outcomes of *Aeromonas* species BSI. This also indicates that *Aeromonas* species are significantly pathogenic and have a high mortality in invasive disease. In a retrospective study of 91 patients with bacteremia in Southern Taiwan, the mortality rate was 23% and active cancer and shock at presentation were independent risk factors for death [20].

*Aeromonas* species can harbor a myriad of intrinsic antimicrobial resistance genes, in particular an array of beta-lactamases and the carbapenemase, CphA [7,9]. It has been proposed that in severe, deep-seated *Aeromonas* infection, that treatment with beta-lactam antibiotics could result in treatment failure and development of resistance [7]. Alternative antibiotic options include fluoroquinolones, cotrimoxazole, aminoglycosides, and cefepime. We demonstrated low rates of resistance to these agents by phenotypic methods as per the CLSI or EUCAST guidelines at the time of reporting in the laboratory.

This population-based, laboratory surveillance, retrospective study had many benefits in its design, however there are some limitations to discuss. We were unable to obtain specific datasets including detailed clinical information, clarification of source in those without a source recorded, surgical procedures or presence of prosthetics or devices, history of water-exposure or injury, or antimicrobial therapy. We were limited by the current laboratory identification and susceptibility testing and reporting at the time of the incident case which varied over the time course of twenty years. We were unable to confirm *Aeromonas* species with genotypic methods or perform further susceptibility testing as the isolates were not available. Therefore, we could not compare between different species including *A. dhakensis*. Studies have suggested that *A. dhakensis* is predominant and more virulent with poorer clinical outcomes [6,33,34,35,36], therefore more investigation is required for this species and comparison to others with outcome data in Australia. Only cases that were admitted to the public system were included and the true rate of *Aeromonas* species BSI in Queensland may be higher than the incidence in this study, albeit small, given private sector cultures were not included in the analysis.

## 5. Conclusions

In this large population study over a two-decade surveillance period in Queensland, Australia, incidence of *Aeromonas* species BSI was 5.2 per million population annually with highest incidence in very remote regions and areas with highest mean annual temperatures. Whilst a rare cause of bacteremia, when present it can cause significant mortality with 30 days case-fatality rates of 19% and significantly higher (28%) with primary bacteremia. The highest risk of death was associated with increasing age, non-focal infection, more comorbidities, and monomicrobial bacteremia. Geographical location, climate zones, and rainfall impacted the occurrence of *Aeromonas* species BSI with higher rates seen in tropical regions with hot, humid and wet summers and increased occurrence with seasonal variation in subtropical areas. However, in regions with high temperatures, *Aeromonas* species BSI occurred sporadically with high population incidence despite low rainfall. With increasing global temperatures and frequency of flooding and natural disasters, we are likely to see increasing rates of *Aeromonas* species BSI. With limited antibiotic options and increasing resistance profiles and with high mortality associated with bacteremia, *Aeromonas* species are an important pathogen to recognize and treat promptly. Further investigation is required to determine the cause of high incidence in very remote regions in Queensland which could lead to preventative measures in the future.

## Figures and Tables

**Figure 1 microorganisms-11-00036-f001:**
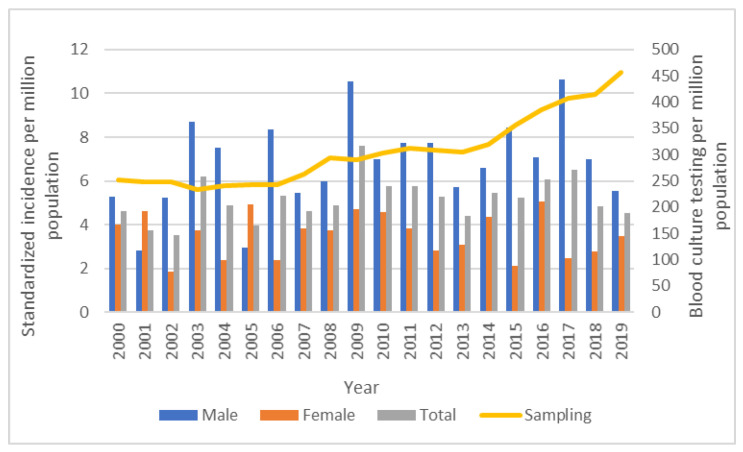
Standardized annual incidence of *Aeromonas* species bloodstream infections in Queensland, Australia, 2000–2019.

**Figure 2 microorganisms-11-00036-f002:**
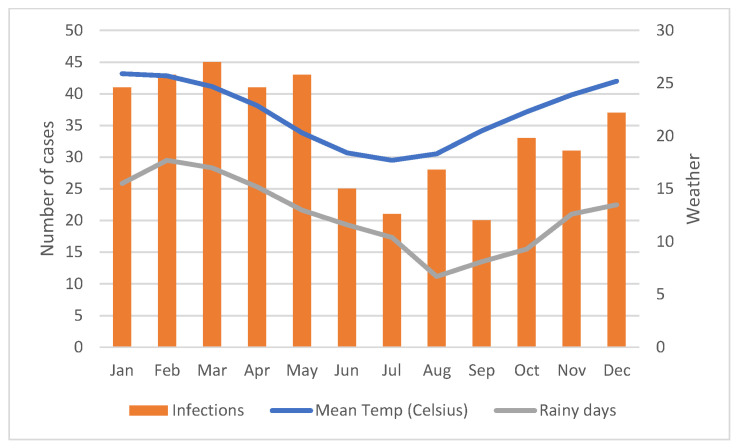
Monthly Aeromonas species bloodstream infections isolated with weather (mean monthly temperature and number of rainy days per month).

**Figure 3 microorganisms-11-00036-f003:**
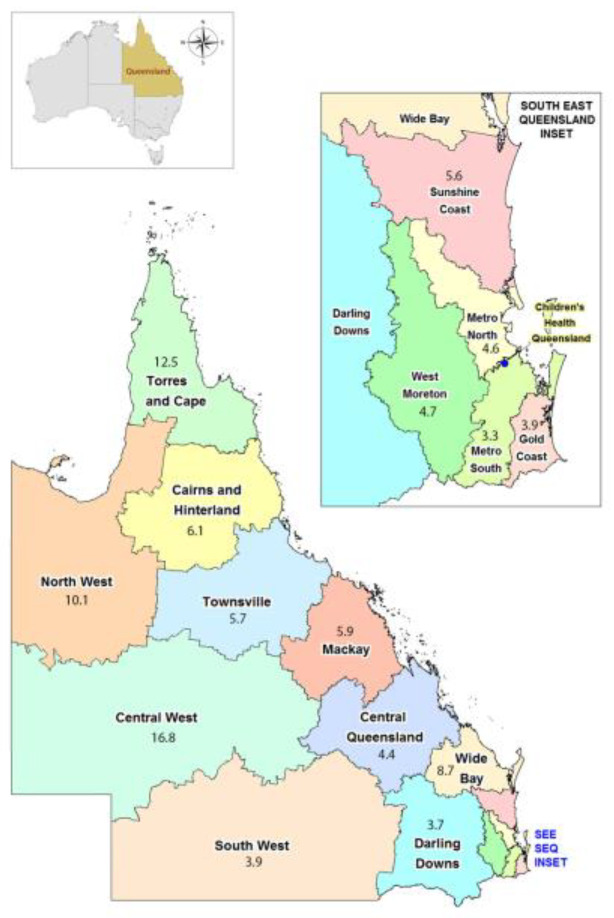
Annual incidence per one million population according to Hospital and Health Service, Queensland.

**Figure 4 microorganisms-11-00036-f004:**
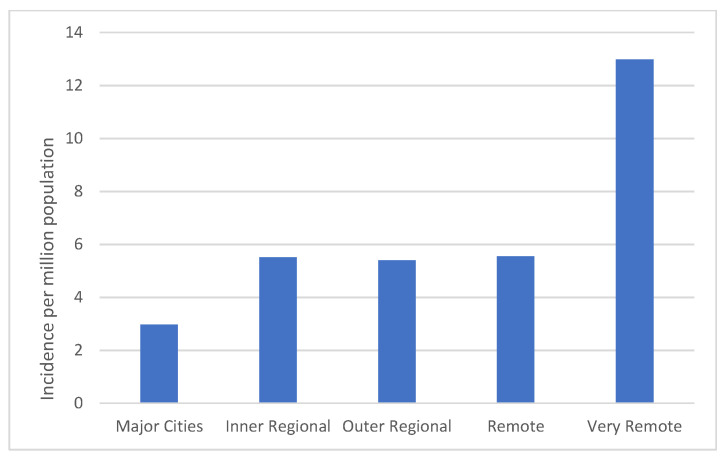
Aeromonas species bloodstream infection incidence per million population by urbanicity in Queensland.

**Figure 5 microorganisms-11-00036-f005:**
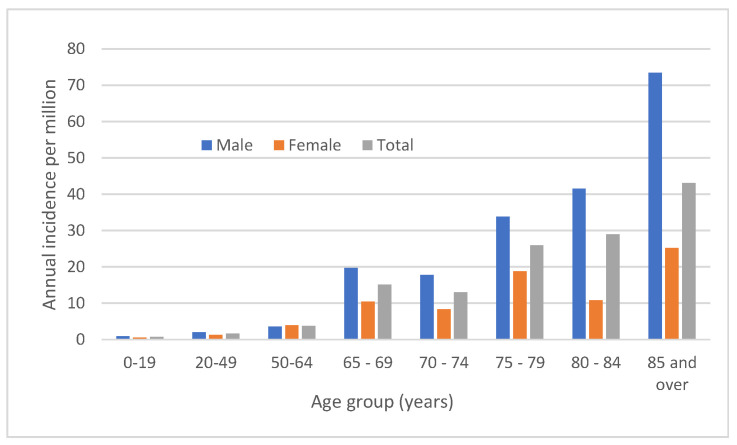
Incidence of Aeromonas species bloodstream infection by age and sex, Queensland, 2000–2019.

**Table 1 microorganisms-11-00036-t001:** Logistic regression modeling of factors associated with 30 days case-fatality in *Aeromonas* species bloodstream infection.

Factor	Odds Ratio	95% Confidence Interval	*p*-Value
Age (per year)	1.03	1.01–1.05	0.001
Charlson comorbidity Index (per point)	1.27	1.15–1.40	<0.001
Polymicrobial	0.36	0.19–0.66	0.001
No focus of infection	2.96	1.69–5.19	<0.001

## Data Availability

Data cannot be shared publicly due to institutional ethics, privacy, and confidentiality regulations. Data release for the purposes of research under Section 280 of the Public Health Act 2005 requires application to the Director General (PHA@health.qld.gov.au).

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
