# Peer review of "Epidemiology of *Aeromonas* Species Bloodstream Infection in Queensland, Australia: Association with Regional and Climate Zones"

_microorganisms, 2022, doi:10.3390/microorganisms11010036_

Round 1

Reviewer 1 Report

The manuscript is well-written and nicely describes the epidemiology of Aeromonas infections. 

The authors should consider commenting on the logistic regression findings, specifically the seemingly "protective effect" of polymicrobial infection shown in Table 1. 

Author Response

The manuscript is well-written and nicely describes the epidemiology of Aeromonas infections.

The authors should consider commenting on the logistic regression findings, specifically the seemingly "protective effect" of polymicrobial infection shown in Table 1.

REPONSE: We do not have an explanation for this finding. We have added additional commentary to the discussion in this regard.

Reviewer 2 Report

In this paper the authors conduct a retrospective study of Aeromonas infections within Queensland over a 20 year period. They report an incidence of just over 5 per million, and their analysis reveals some co-morbidity factors associated with infections. Generally the paper is well written and the conclusions are justified. A short section at the end also points out the limitations of the study.

I have only a couple of suggestions to improve the manuscript.

1.     Page 2, last line of introduction. I don’t think “secular” is the right word to use here? What is the meaning the authors are trying to convey- geographical or regional, or something else? To me secular means non-religious.

2.     Figure 4 legend: “Figure 4. Aeromonas species bloodstream infection incidence per million population and remoteness areas in Queensland” What is meant by “remoteness areas”? I think this should be rephrased for clarity.

3.     Figure 5- for consistency with figure 1, change F M T, to female, male and total incidence in figure key. Also- the colours have changed from figure 1- I’d suggest keeping it consistent across the two figures: male is blue in figure 1 but red in figure 2, and vice versa for female.

4.     Section 3.3 “The median age…” insert “for Aeromonas infection was…”

Author Response

Reviewer 2.

In this paper the authors conduct a retrospective study of Aeromonas infections within Queensland over a 20 year period. They report an incidence of just over 5 per million, and their analysis reveals some co-morbidity factors associated with infections. Generally the paper is well written and the conclusions are justified. A short section at the end also points out the limitations of the study.

I have only a couple of suggestions to improve the manuscript.

  1. Page 2, last line of introduction. I don’t think “secular” is the right word to use here? What is the meaning the authors are trying to convey- geographical or regional, or something else? To me secular means non-religious.

RESPONSE: The use of the term secular trends is common in the literature. A pubmed search of “secular” and “bloodstream” returns more than 50 publications, many using this terminology in the title. We have further reworded to “secular trends”. However, if this remains unsatisfactory the word secular could be switched to “temporal”.

  1. Figure 4 legend: “Figure 4. Aeromonas species bloodstream infection incidence per million population and remoteness areas in Queensland” What is meant by “remoteness areas”? I think this should be rephrased for clarity.

RESPONSE: We have changed to “urbanicity” in the legend (defined as the number of people relative to area surface according to one definition https://www.epa.gov/caddis-vol2/urbanization-overview).

  1. Figure 5- for consistency with figure 1, change F M T, to female, male and total incidence in figure key. Also- the colours have changed from figure 1- I’d suggest keeping it consistent across the two figures: male is blue in figure 1 but red in figure 2, and vice versa for female.

RESPONSE: The Figure 5 male and female order and colours have been changed for consistency in presentation as recommended.

  1. Section 3.3 “The median age…” insert “for Aeromonas infection was…”

RESPONSE: Revised as requested.

Reviewer 3 Report

Dear Authors,

The manuscript ID: microorganisms-2114766_v1 entitled Epidemiology of Aeromonas species bloodstream infection in Queensland, Australia: Association with regional and climate zones” is very interesting. The purpose of the work is concrete and very original from the epidemiological point of view of Aeromonas species bloodstream infection. The whole article (Introduction, Materials and Methods, Results, Discussion and Conclusions) is properly organized. The introduction is concise and provides general data on Aeromonas species. Appropriate methods were used to perform these studies. Statistical analysis was also performed. Results are documented, presented in the form of figures and tables and interpreted. There is also an interesting and extensive discussion. Based on the results, some conclusions were drawn. The Authors confirmed that Aeromonas species are an important pathogen that needs to be recognized and treated quickly. I think, it is a well written manuscript.

I have only small suggestions for the paper, which are as follows:

1)   Figure 2 could be clearer for the reader. Does the range of values 0 - 50 mean both the number of strains and the average temperature?

2)   The names of microorganisms throughout the text should be written in italics.

According to me, this retrospective population-based cohort study is very valuable and worth publishing in “Microorganisms”.

With highest regards,

Author Response

I have only small suggestions for the paper, which are as follows:

1)   Figure 2 could be clearer for the reader. Does the range of values 0 - 50 mean both the number of strains and the average temperature?

RESPONSE: We have revised the title, axes, and labels to further clarify the figure contents.

2)   The names of microorganisms throughout the text should be written in italics.

RESPONSE: Revised.

According to me, this retrospective population-based cohort study is very valuable and worth publishing in “Microorganisms”.